# A Disjunctive Marginal Edge of Evergreen Broad-Leaved Oak (*Quercus gilva*) in East Asia: The High Genetic Distinctiveness and Unusual Diversity of Jeju Island Populations and Insight into a Massive, Independent Postglacial Colonization

**DOI:** 10.3390/genes11101114

**Published:** 2020-09-23

**Authors:** Eun-Kyeong Han, Won-Bum Cho, Jong-Soo Park, In-Su Choi, Myounghai Kwak, Bo-Yun Kim, Jung-Hyun Lee

**Affiliations:** 1Department of Biological Sciences and Biotechnology, Chonnam National University, Gwangju 61186, Korea; urinara-han@hanmail.net; 2Department of Biology Education, Chonnam National University, Gwangju 61186, Korea; rudis99@hanmail.net; 3Department of Biological Sciences, Inha University, 100, Inha-ro, Michuhol-gu, Incheon 22212, Korea; beeul25@gmail.com; 4School of Life Sciences, Arizona State University, Tempe, AZ 85287, USA; 86ischoi@gmail.com; 5Biological and Genetic Resources Utilization Division, National Institute of Biological Resources, Incheon 22689, Korea; mhkwak1@korea.kr; 6Plant Resources Division, National Institute of Biological Resources, Incheon 22689, Korea; bykim416@korea.kr

**Keywords:** marginal edge, *Quercus gilva*, genetic diversity, massive colonization, Jeju Island, conservation

## Abstract

Jeju Island is located at a marginal edge of the distributional range of East Asian evergreen broad-leaved forests. The low genetic diversity of such edge populations is predicted to have resulted from genetic drift and reduced gene flow when compared to core populations. To test this hypothesis, we examined the levels of genetic diversity of marginal-edge populations of *Quercus gilva*, restricted to a few habitats on Jeju Island, and compared them with the southern Kyushu populations. We also evaluated their evolutionary potential and conservation value. The genetic diversity and structure were analyzed using 40 polymorphic microsatellite markers developed in this study. Ecological Niche Modeling (ENM) has been employed to develop our insights, which can be inferred from historical distribution changes. Contrary to our expectations, we detected a similar level of genetic diversity in the Jeju populations, comparable to that of the southern Kyushu populations, which have been regarded as long-term glacial refugia with a high genetic variability of East Asian evergreen trees. We found no signatures of recent bottlenecks in the Jeju populations. The results of STRUCTURE, neighbor-joining phylogeny, and Principal Coordinate Analysis (PCoA) with a significant barrier clearly demonstrated that the Jeju and Kyushu regions are genetically distinct. However, ENM showed that the probability value for the distribution of the trees on Jeju Island during the Last Glacial Maximum (LGM) converge was zero. In consideration of these results, we hypothesize that independent massive postglacial colonization from a separate large genetic source, other than Kyushu, could have led to the current genetic diversity of Jeju Island. Therefore, we suggest that the Jeju populations deserve to be separately managed and designated as a level of management unit (MU). These findings improve our understanding of the paleovegetation of East Asian evergreen forests, and the microevolution of oaks.

## 1. Introduction

It is well known that the population genetic structure in extant plants is affected by various factors, including the dispersal ability of pollinators, seed dispersal modes, reproductive systems, and historical migration patterns; the historical range change during Quaternary climatic oscillations is also considered a primary factor [1,2,3]. Although their relative importance may vary across time and space, the genetic features of populations in East Asian temperate regions likely reflect historical, rather than current, levels of gene flow [4,5,6,7,8]. East Asia has experienced complex and dynamic changes in land configurations during the Quaternary period, which led to a high richness and endemism of plant species in forests [9]. The range change of warm temperate evergreen forests was larger than that of temperate deciduous forests, especially in Korea and Japan.

Peripheral, especially marginal/edge, populations, might reflect genetic impoverishment as a result of genetic drift and reduced gene flow when compared to core populations [10,11,12,13]. Such genetic determinants have the potential to further expand species ranges through adaptation to the selection pressures of a marginal environment, assumed to furnish less fitness for their survival [14]. The populations have played decisive roles for species facing and responding to rapidly changing environmental conditions [15,16]. Many of these studies of local fitness have improved our knowledge of how a given species adapts to a changing environment [17,18]. Nonetheless, our understanding of the evolution of species in warm temperate evergreen forests in East Asia is still lacking.

The volcanic Jeju Island of South Korea is characterized by its high endemism, unique altitudinal zonation of vegetation, and untouched environments [19], and thus it was designated as a UNESCO Biosphere Reserve in 2002 and a World Heritage Site in 2007. The island is disjunctively located at the marginal edge of the distributional range of East Asian evergreen broad-leaved forests. The lowland zone of Jeju Island is covered with forests, dominated by warm temperate and subtropical evergreen broad-leaved plants [20]. These species commonly inhabit large ranges across East Asia, including South Korea, Japan, and China, but exhibit disjunctive distributions, with a heterogeneous boundary of habitat preferences [21]. Therefore, conserving the populations at the marginal edge of their range can be beneficial to the long-term survival of a species.

*Quercus gilva* Blume (Fagaceae) is a large, ecologically important tree of evergreen broad-leaved forests in East Asia [22]. Although *Q. gilva* is widely distributed in East Asian forests [23,24], its habitat is decreasing due to anthropogenic pressure. The main cause of habitat decrease is human-mediated disturbance, such as large-scale regional development [25] and logging [26]. The population of Jeju Island is extremely small (total: ca. 600 individuals; [27]) and distributed in a unique habitat, called Gotjawal, where the trees occur in an area made up of numerous fragmented rocks. This species is listed as Vulnerable (VU) in the Korea Red Data Book [28]. As of 2012, it has also been protected under the Endangered Species Act (ESA) within Korean law. Since tracking of intraspecific Conservation Units (CUs) is one of the most important tasks for the long-term conservation of a given species, a population genetic examination for *Q. gilva* was attempted using RAPD (Random Amplified Polymorphic DNA) [29] and ISSR (Inter Simple Sequence Repeat) [25] analysis. However, previous assessments have not been performed with other comparative populations, which could be a criterion for accurately recognizing their genetic status. From a recent conservation genetics point of view, providing information on the population’s establishment history is becoming a fundamental step in long-term conservation [30,31,32].

Fossils and pollen grains could be utilized for unraveling evolutionary clues that contribute to present genetic diversity, but the situation is complicated by historically complex distribution changes [33,34]. However, such past data are largely absent in East Asia, because the area in which the evergreen forests appear to have been historically distributed is now in the sea. Given this, a technique such as Ecological Niche Modeling (ENM) is the only auxiliary way to reveal the historical distribution of a species [35]. ENM is useful in genetic studies to infer climate change-associated correlations between distribution shifts and genetic structure [36,37,38].

As has been observed in other warm temperate species in East Asia [5,6,39], the extant Jeju populations of *Q. gilva* are most closely related to those in Kyushu, Japan, which is geographically adjacent and has a similar establishment history. Therefore, we characterized the genetic compositions of the *Q. gilva* populations in Jeju Island, located at the disjunctive edge of their distribution range. The genetic diversity was compared to the Kyushu populations, regarded as long-term glacial refugia with a high genetic variability of East Asian warm temperate evergreen broad-leaved trees. The purposes of the present study are (1) to develop a high-resolution and cost-effective polymorphic microsatellite set so that researchers can continue periodic genetic monitoring, (2) to evaluate the evolutionary potential and conservation value of marginal-edge Jeju populations by inferring the history of population establishment, and (3) to provide conservation guidelines for the recovery and management of the threatened Jeju populations. The genetic diversity and structure were analyzed using 40 polymorphic microsatellite markers developed in this study through high-throughput sequencing data. ENM was also employed to examine historical distribution changes.

## 2. Materials and Methods

### 2.1. Plant Material Sampling and DNA Extraction

We collected a total of 158 leaf samples of *Q. gilva* from three populations in Jeju Island, Korea and three populations in Kyushu, Japan. Since *Q. gilva* is protected as an endangered species in Korea, we first requested permission from the Ministry of Environment and then proceeded with the material collection. We selected the trees with a diameter at breast height (DBH) of more than 20 cm while maintaining minimal intervals of more than 5 m between individuals. One leaf sample was collected per individual to minimize damage to the species. In Jeju populations, a total of 77 leaves, including 32 from Gueok-ri (k-GU), 27 from Jeoji-ri (k-JJ), and 18 from Seogwang-ri (k-SG), were obtained, with an average of 25.6. In the Kyushu populations, an average of 27 and total of 81 leaf samples were collected from Kitadake, Kumamoto Prefecture (j-GM; 23), and Aoidake (j-MY; 29) and Enodake (j-NB; 29) in Miyazaki Prefecture. Collected leaf samples were stored at −80 °C in a deep freezer at the lab of Biological Education, Chonnam National University until use. Total genomic DNA was extracted from dried leaf samples using the DNeasy Plant mini kit (Qiagen, Seoul, Korea) following the manufacturer’s instructions. The concentration of extracted DNA was determined using Nano-300 (Allsheng, Hangzhou, China), and diluted to 15 ng/μL to obtain the same concentration of template DNA in each sample.

### 2.2. Loci Isolation for Microsatellite Markers’ Development and Genotyping

In order to develop polymorphic microsatellite markers for *Q. gilva*, we produced high-throughput sequencing data in a fresh leaf collected from Gueok-ri, Seogwipo-si, JeJu Island, Korea. A voucher specimen was deposited in the herbarium of Chonnam National University (BEC) (Voucher no. LeeQg20180502). A shotgun library construction for DNA sequencing was generated using the Illumina MiSeq platform (LAS, Seoul, Korea). According to the method of Cho et al. [40], we detected di-, tri-, or tetranucleotide motifs with flanking regions >100 bp and at least 10, six, or four repeats, respectively, through SSR_pipeline v. 0.951 [41]. After acquiring reads containing microsatellites from this screening, we attempted a reference mapping of the total paired reads to each remaining sequence using Geneious R11.0.5 [42]. In the reference-mapped results, after discarding putative multicopy loci with exceptionally high coverage (>20 reads), we used the final reads, showing the variation in length at the repeating site, no substitution of the site to produce the primer, and no additional insertion/deletion in the flanking region. Based on the final selected reads, we designed 54 primer pairs using Primer3 version 0.4.0 software [43] in the Geneious program according to the following parameters: primer size 18–22 bp, Tm (melting temperature) of 53–60 °C, and GC content of 35–65%. The forward primers added three sets of M13 tag sequences (5′-CACGACGTTGTAAACGAC-3′, 5′-TGTGGAATT GTGAGCGG-3′, and 5′-CTATAGGGCACGCGTGGT-3′) with 6-FAM, VIC, and NED fluorescent dye, respectively.

To assess the polymorphisms for the designed microsatellite loci, we conducted a preliminary PCR analysis with 32 individuals from the Gueok-ri population. PCR amplification was performed with a Veriti 96-well thermal cycler (Applied Biosystems, Foster City, CA, USA) using 5 μL volumes that were composed of 15 ng of extracted DNA, 2.5 μL Multiplex PCR Master Mix (Qiagen, Valencia, CA, USA), 0.01 μM forward primer, 0.2 μM reverse primer, and 0.1 μM of the M13 primer (fluorescently labeled). PCR amplification was performed as follows: initial denaturation at 95 °C for 15 min; 35 cycles of denaturation at 95 °C for 30 s, annealing at 56 °C for 1.5 min, and extension at 72 °C for 1 min; a final extension at 72 °C for 10 min. The PCR products were diluted at 1:30, and 1 uL was analyzed on an ABI 3730XL sequencer with GeneScan^TM^-500LIZ^TM^ Size Standard (Applied Biosystems). Allele sizes and peaks for each sample were determined three times via Peak Scanner software 2 to minimize genotyping errors. We selected 46 polymorphic microsatellite loci with clear, strong peaks for each individual. Then, we tested the remaining 126 individuals from five populations according to the DNA extraction and PCR protocols described above.

### 2.3. Statistical Data Analysis

Before inferring the genotyping data, we estimated the null allele frequency using INEst (inbreeding/null allele estimation) software based on the individual inbreeding model (IIM), which calculates the null allele frequency regardless of the effect of inbreeding [44]. This analysis showed that six loci (Qrg009, Qrg013, Qrg026, Qrg030, Qrg036, and Qrg048) showed a null allele frequency of more than 5%. Therefore, we used a total of 40 microsatellite markers, except for the six loci, for statistical analysis.

The summary genetics statistics were calculated at the population and pooled regional population levels. These included the number of alleles (*N*_A_), the number of private alleles (*P*_A_), the private allele rate (*P*_riv_), the mean expected heterozygosity (*H*_E_), the mean observed heterozygosity (*H*_O_), and the fixation index (*F*_IS_), calculated using GenAlEx 6.5 [45]. The allele richness (*A*_R_) and genetic differentiation among populations (*F*_ST_) were determined by calculating the overall *F*_IS_ according to the method of Weir and Cockerham [46], using FSTAT 1.2 [47]. The statistical significance of *F*_ST_ was tested using the log-likelihood (*G*)-based exact test in FSTAT. To test for departures from Hardy–Weinberg equilibrium (HWE) and linkage equilibrium, we conducted exact tests based on a Markov chain method (1000 permutations), using GENEPOP 4.0 [48]. The possibility of recent bottleneck for population was detected using BOTTLENECK 1.2.02 [49] (1000 iterations). We utilized two models for evolution—a two-phase model (TPM; the proportion of the stepwise mutation model (SMM) in TPM = 0.000, variance of the geometric distribution for TPM = 0.36), and a stepwise mutation model (SMM)—in a BOTTLENECK analysis that included the Bayesian Wilcoxon signed-rank test, to evaluate departures from a 1:1 deficiency/excess ratio [50]. The possibility of population bottleneck was also estimated by a mode-shift test, which detects disruptions in the distribution of allelic frequencies [50].

To analyze the population structure, we used a Bayesian clustering approach implemented in STRUCTURE 2.3, as calculated from microsatellite markers [51], using 1,000,000 Markov Chain Monte Carlo (MCMC) iterations (100,000 burn-in, with admixture). The simulation used 20 iterations, with *K* = 1 to *K* = 7 clusters. The optimal number of clusters, *K*, was found via the *K* method, using STRUCTURE HARVESTER [52]. CLUMPP v. 1.1.2 [53] with the Greedy algorithm was used to combine the membership coefficient matrices (*Q*-matrices) from 1000 iterations for *K* = 2, using random input orders.

To test for the presence of isolation-by-distance (IBD), we used Mantel tests in GenAlEx 6.5 [45] with 999 random permutations; this requires a correlation analysis between the pairwise *F*_ST_ values, and measurements of geographic distance between populations. To identify genetic boundaries between populations, we performed a barrier analysis [54] based on Monmonier’s algorithm [55] with 1000 bootstrap matrices of pairwise *D*_A_ standard genetic distance [56] that were calculated by MICROSATELLITE ANALYZER (MSA) v. 4.05 [57]. The distance matrices were also used to construct a 50% consensus tree by the Neighbor-Joining (N-J) method, as implemented in PHYLIP v. 3.68 [58]. To find the genetic structure of *Q. gilva*, a principal coordinate analysis (PCoA) was conducted by the covariance standardized approach of pairwise Nei’s genetic distances in GenAlEx 6.5.

### 2.4. Ecological Niche Modeling

We modeled the present and past (during LGM) potential distributions of *Quercus gilva* using Maxent 3.4.1 [59]. Occurrence data for this species included sample localities from our study as well as published data [27] and GBIF data with preserved specimens [60]. We obtained 242 occurrence data points and the occurrence data were spatially rarefied using SDMtoolbox 2.4 [61] to reduce bias in developing the distribution model. Two occurrence points of Korean Peninsula (inland) and Toyama Prefecture in Japan were excluded because they were estimated to be distributed in uncertain and inappropriate climate zones. A total of 97 occurrence data points were finally used in ENM. We obtained 19 bioclimatic variables (Online Resource 2) for the present and LGM from Climatologies at High Resolution for the Earth’s Land Surface Areas (CHELSA, http://chelsa-climate.org/; [62]). We obtained elevation data for the present—the Global Multi-resolution Terrain Elevation Data (GMTED2010) dataset [63]—from the USGS EROS Archive (https://www.usgs.gov/land-resources/eros/coastal-changes-and-impacts/gmted2010), and for the LGM from CHELSA. To reconstruct the historical distributions, we utilized three past climate models for LGM: the Community Climate System Model (CCSM4; [64]), the Earth System Model based on the Model for Interdisciplinary Research on Climate (MIROC-ESM; [65]), and the Max Planck Institute for Meteorology Earth System model (MPI-ESM-P). We selected one of the climate variables and elevation data sharing a high Spearman correlation efficient (>0.7) by using SDMtoolbox 2.4 [61], in order to avoid multicollinearity problems. Therefore, 7 of 20 variables were used in ENM. To reduce the effects of uncertainty in the historical climate models, we averaged the historical distributions that were based on each of the three climate models. The climate data, for 20–37° N and 115–145° E (30 arcsecond resolution), were extracted using ArcGIS 10.5 (ESRI 2017). Maxent runs were performed in batch mode with these settings: create response curves, conduct jackknife tests, use 20 replicates, generate logistic output, select random seeds, and we used 10,000 background points and 1,000 iterations.

## 3. Results

### 3.1. Development of Polymorphic Microsatellite Markers

In total, 11,957,206 reads were generated by Illumina paired-end sequencing (Short Read Archive accession number: PRJNA649602). The total number of reads containing microsatellites identified through the SSR-pipeline was 100,849 reads, including 55,084 reads with dinucleotide motifs, 41,037 reads with trinucleotide motifs, and 4,728 reads with tetranucleotide motifs. Of these, the di-, tri-, and tetranucleotide motifs with planking areas of >100 bp and having repeating units of at least 12, 6, and 6, respectively, were 29,058 reads, 21,424 reads, and 2,562 reads.

As a result of applying 54 designed microsatellite loci to 32 individuals of *Q. gilva* from Gueok-ri populations in Korea, 46 polymorphic microsatellite markers with clear and strong peaks for each allele were selected (Table 1). Regarding the results of the genetic diversity analysis, a total of 385 alleles were detected in 46 microsatellite loci across all samples. The number of alleles (*N*_A_) per locus ranged from 2 to 19, with an average of 8.370 alleles per locus. Values for observed heterozygosity (*H*_O_) and expected heterozygosity (*H*_E_) ranged from 0.044 to 0.918 (mean: 0.616) and from 0.067 to 0.899 (mean: 0.664), respectively (Table 2). The inbreeding coefficient (*F*_IS_) for each locus ranged from −0.135 to 0.669. The null allele frequency identified by INEst software ranged from 0.0018 to 0.2774. Comparing the genetic diversity (*N*_A_ and *H*_E_) by the locus of di-, tri-, and tetranucleotide motifs, the results were higher in loci with dinucleotide motifs (mean *N*_A_ = 11.125, *H*_E_ = 0.784) than with tri- (mean *N*_A_ = 5.571, *H*_E_ = 0.527) or tetranucleotide motifs (mean *N*_A_ = 5.000, *H*_E_ = 0.547) (Figure 1).

All the developed markers were deposited in in the National Center for Biotechnology Information’s GenBank database (Table 1). As a result, a cost-effective set of 46 polymorphic microsatellite markers with high resolutions has been successfully developed. These markers will be useful for conserving genetic resources through periodical monitoring management, creating a seed genogram, cloning detection, and postrestoration assessments in the endangered Jeju populations of *Q. gilva*.

### 3.2. Genetic Diversity

Genetic diversity parameters, evaluated at the population and pooled regional population levels for all 158 individuals of *Q. gilva*, are shown in Table 3. In total, 335 alleles were amplified from 40 microsatellite loci, with an average of 8.4 alleles per locus. They ranged from a minimum of two (Qrg025) to a maximum of 19 (Qrg021). Among the six *Q. gilva* populations, the levels of genetic diversity showed no noticeable difference; the number of alleles ranged from 224 to 261 (mean of 244.7); *H*_E_ ranged from 0.615 (j-NB) to 0.651 (k-SG) (mean of 0.639); *A*_R_ ranged from 5.201 (j-NB) to 5.940 (j-MY) (mean of 5.726); *P*_A_ ranged from 4 (k-JJ and j-NB) to 11 (k-GU) (mean of 7); *F*_IS_ ranged from −0.024 (k-GU) to 0.043 (k-SG) (mean of 0.007). The k-SG population had the highest genetic diversity values, while the j-NB population had the lowest (*H*_E_, *A*_R_). In the comparison between the Jeju and Kyushu populations, the levels of genetic diversity were almost equivalent, showing only a slight difference in degree depending on the parameters (Table 3). The BOTTLENECK analysis (Wilcoxon tests) showed no significant bottleneck effects across all populations under a TPM and SMM (*p* > 0.05), as well as a mode shift (Table 4).

### 3.3. Population Structure

To infer the population structure of *Q. gilva*, we performed STRUCTURE, N-J phylogeny, and PCoA analysis with 40 microsatellite loci. The results clearly demonstrated that the Jeju and Kyushu regions are genetically distinct. The STRUCTURE analysis showed that the optimal *K*-value was 2 for Δ*K* = 199.678 and the second fit value was 4 for Δ*K* = 6.319. At *K* = 2, a strong genetic structure was found among populations, divided clearly into two regions (Figure 2 and Figure 3). In terms of neighbor-joining criteria, the sampled populations of *Q. gilva* were clearly divided into two clusters (Jeju and Kyushu), in concordance with the clustering results obtained by STRUCTURE (Figure 4). The principal coordinate analysis (PCoA) results revealed a population structure that was in accordance with the STRUCTURE and N-J phylogeny analysis (Figure 5). The first two coordinates explained 7.32% (4.12% for axis 1 and 3.20% for axis 2) of the total genetic variation. Based on pairwise *F*_ST_, the barrier analyses identified a strong barrier between the Jeju and Kyushu regions (Figure 2). Although the *F*_ST_ values of pairwise comparisons among the six populations showed a numerically low overall genetic differentiation, with a *F*_ST_ of 0.029 with 95 and 99% confidence intervals of 0.024–0.034 and 0.022–0.036, respectively, it was significant at all loci (*p* < 0.001). This differentiation was also seen between populations within Jeju Island (mean 0.010, with 95 and 99% confidence intervals of 0.005–0.017 and 0.003–0.019, *p* < 0.001), and within populations of Kyushu (mean 0.021, with 95 and 99% confidence intervals of 0.015–0.027 and 0.013–0.029, *p* < 0.001). Therefore, the fact that the highest value of genetic differentiation is from the overall population means that it results from a difference between Jeju Island and Kyushu (Figure 6b). Additionally, isolation by distance (IBD) analysis, as determined by Mantel test, showed a significant correlation between genetic and geographic distance among populations (*R*^2^ = 0.7611, *p* = 0.025) (Figure 6a).

### 3.4. Ecological Niche Modeling

The ENM of *Q. gilva* (Figure 7) had a high average AUC (area under the curve) (0.899), supporting its predictive power. The most important variable was bio_02 (mean diurnal range; 55.8%), followed by bio_12 (annual precipitation; 14.6%) and bio_15 (precipitation seasonality; 10.9%). The estimated LGM distribution was near the paleo-coastline with no inland potential distribution (Figure 7b). The potential value of more than 0.500 were shown in southern Kyushu, the central East China Sea, southeastern Taiwan, and the Ryukyu archipelago (Figure 7b).

## 4. Discussion

One of the most notable results of this study is that the geographically adjacent Jeju and Kyushu populations are genetically divergent. Comparison with other evergreen trees belonging to the same forest biomes may help to explain such a genetic pattern. Although the available data for genetic examination in this region are still minimal, consistent results point toward the fact that the warm temperate evergreen broad-leaved trees of South Korea, including the Jeju populations, have been affected by postglacial migration from those of Kyushu, Japan (*Neolitsea sericea*: [5]; *Machilus thunbergii*: [39]; *Quercus acuta*: [6]). Previous studies have shown that the Korean populations are homogeneous, with a genetic structure that is not very distinct from those of Kyushu, Japan. Furthermore, *Q. gilva* has an almost identical life history to *Q. acuta*; it is wind-pollinated, and the nuts contain a one-seeded fruit with a hard wall that is usually dispersed by small rodents such as squirrels and jays or animal-cached [70,71]. Therefore, the contrasting pattern of genetic structure suggest that historical factors are the most relevant. Therefore, we suggest that *Q. gilva* has a unique and separate evolutionary history.

Regardless of the latitude, the distribution of warm temperate evergreen broad-leaved forests in East Asia, such as in Korea, China, and Japan, is clearly related to the flow of the Kuroshio warm current (KC), showing a unique distribution structure [34,72,73]. Therefore, despite being located at relatively high latitudes, the southwestern portions of the Japanese main islands (i.e., southern areas of Kyushu, Shikoku, and Honshu) have long been regarded as crucial refugia with high genetic variation for evergreen broad-leaved trees that resulted from southward range shifts that paralleled glacial cycles [74,75,76]. ENM also revealed that *Q. gilva* populations are distributed like shadows along the flow of the present KC as well as the LGM (Figure 7). Therefore, considering the geographical location and population size of Jeju Island, it is worth noting that the levels of genetic diversity between the Jeju and Kyushu regions are almost equally similar in both populations and pooled regional populations. We found that the Jeju populations harbor a level of allele richness (*A*_R_) and private allele rate (*P*_riv_) comparable to those of Kyushu populations. Although, such indicators provide a strong possibility for the existence of refugia in this area [77,78,79], we are convinced that Jeju Island did not serve as the glacial refuge of *Q. gilva*. In agreement with our assumption, ENM showed that the probability values for the distribution of the trees on Jeju Island during the LGM converged at zero. East Asian evergreen oaks prefer different habitats depending on the elevation gradient along the mountain slopes. A previous study suggested that Jeju Island was the refugia of *Q. acuta* [6], which occurs at the highest elevation range [80]. By comparison, *Q. gilva* shows a bias towards low-elevation stands, generally forming a community with *Q. glauca* [80]. In fact, *Q. acuta* occupies the middle of Mt. Halla on Jeju Island (approximately 600 m a.s.l.), while *Q. gilva* is distributed very close to the southern coastline (average: 168 m a.s.l.). It should be taken into account that the Quaternary climate oscillations caused not only latitudinal changes in the distribution of a given species but also vertical elevation migration [38]. During the glacial periods, in addition to the cold climate, competition for vertical migration with other oak species would have prompted the Jeju populations to retreat to low latitudes. The ENM revealed that, due to an apparent range contraction southward along the paleo KC, with the exception of the Ryukyu Archipelago, the high distribution potentials have been narrowed to three places—southeastern Taiwan, central East China Sea, and southern Kyushu in LGM (Figure 7b). Given that the Jeju Island populations are genetically different from Kyushu’s, the glacial refugia (i.e., genetic source) of the current Jeju populations would have most likely existed somewhere around the central East China Sea, which was land during the LGM.

One plausible scenario might be hypothesized for the unusual level of genetic diversity in Jeju Island, i.e., a massive postglacial immigration, demographically independent of Kyushu. Most oak species have common ecological traits, but their distinct adaptability also facilitates domination in such areas. In fact, evergreen *Quercus* species frequently dominate the landscape in extensive regions of East Asia, because they are better able to form dense and crowded stands [81,82]. In general, the larger the population is, the more reduced the effect of genetic drift is, which promotes a reduction in genetic diversity. We found no indication of significant recent bottlenecks, implying that the founding populations of *Q. gilva* might have been large enough to weaken the effect of genetic drift. Therefore, we suggest that a massive postglacial colonization, which maintained a high genetic diversity from separately stable and large genetic sources other than Kyushu, could have led to the current genetic diversity of Jeju Island. Further work using additional samples, including from broad areas such as the Ryukyu Islands and Eastern China, and the markers developed in this study might provide a better understanding of the historical migration of the warm temperate evergreen tree *Q. gilva* in Jeju Island. In particular, how the Jeju populations relate to East Asian populations other than Kyushu populations should be tested.

There are several definitions for determining the levels of CUs, such as Evolutionarily Significant Units (ESUs) and Management Units (MUs), because “high divergence” is too vague a term for practical purposes [83,84,85]. However, the criteria of the MUs clearly represent demographically independent units that merit separate management [86,87]. Considering the high genetic distinctiveness with a significant barrier, monophyletic phylogeny, population size, and other evidence, we suggest that the Jeju populations should be separately managed as a MU. The notable and unique genetic diversity of Jeju populations represents a high value in terms of conservation as it can contribute to the species’ genetic diversity. Such genetic determinants should be well preserved and returned when East Asian populations are reconnected in response to the climate fluctuation. The low degree of genetic differentiation among the populations within Jeju Island suggests that all populations should be integrated and managed together rather than focusing conservation efforts on any particular subset of the population. From a long-term conservation genetics perspective, it is especially important for *Q. gilva* that conservation efforts should be focused on prohibiting the large-scale industrial development of the habitat, because large trees are not vulnerable to personal interference, such as overcollection. Thus, first, we recommend that all known habitats be protected in situ by law to prevent further damage. To prepare for inevitable land development, we suggest that ex situ preservation should be preceded by efforts to store good-quality seeds. Finally, if the artificial restoration of habitats is required, note that the source for the Jeju populations is not Kyushu.

## Figures and Tables

**Figure 1 genes-11-01114-f001:**
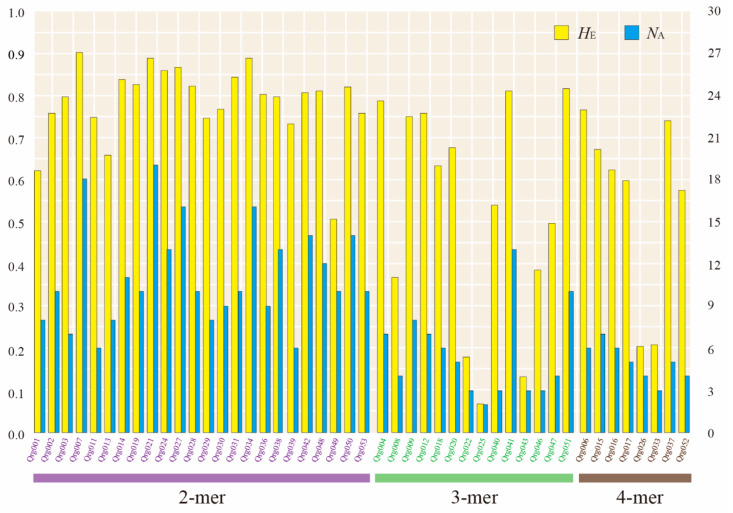
Comparing genetic diversity (*N*_A_ and *H*_E_) by locus with di-, tri-, and tetranucleotide motifs. Genetic diversity is based on the allele frequency of six populations of *Quercus gilva* using 46 microsatellite loci.

**Figure 2 genes-11-01114-f002:**
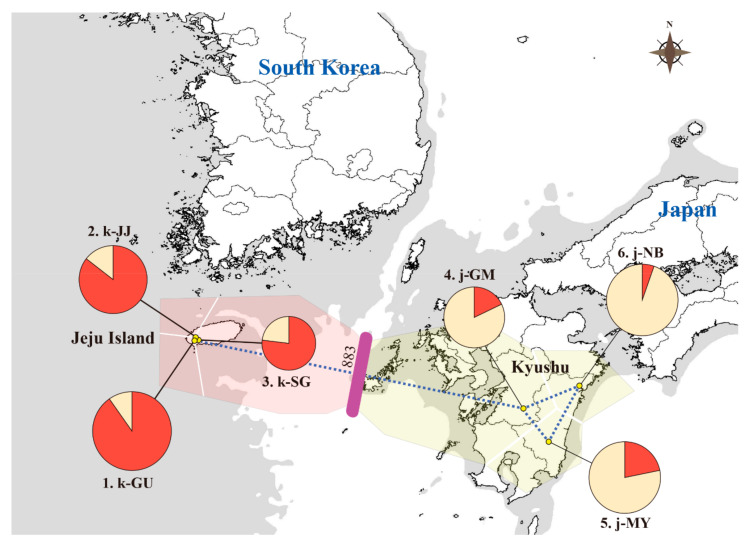
Genetic composition and a genetic barrier of *Quercus gilva* geographic populations using 40 microsatellite loci. Genetic composition is based on STRUCTURE clustering results (*K* = 2). The genetic barrier is marked with a thick purple line, estimated by BARRIER. The gray shading represents exposed coastal areas and sea basins during times of glacially induced alterations in sea levels during the Late Pleistocene.

**Figure 3 genes-11-01114-f003:**
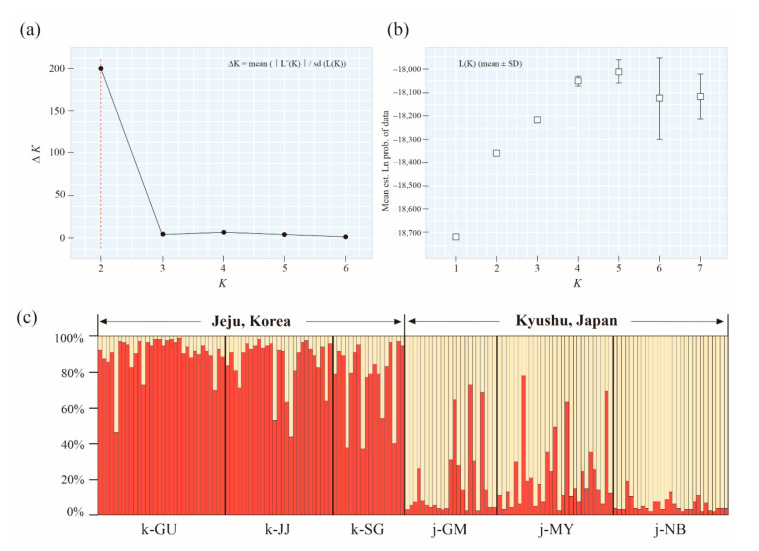
Plots generated in STRUCTURE Harvester (**a**) Evanno’s delta *K* statistic; (**b**) the mean log-likelihood of the data L(K). Genetic structure of *Quercus gilva* populations based on Bayesian assignment tests performed in STRUCTURE. (**c**) Genetic structural plot of *Q. gilva* populations at *K* = 2. Each individual is represented by a single vertical line that represents the individual’s estimated membership fractions in these two clusters.

**Figure 4 genes-11-01114-f004:**
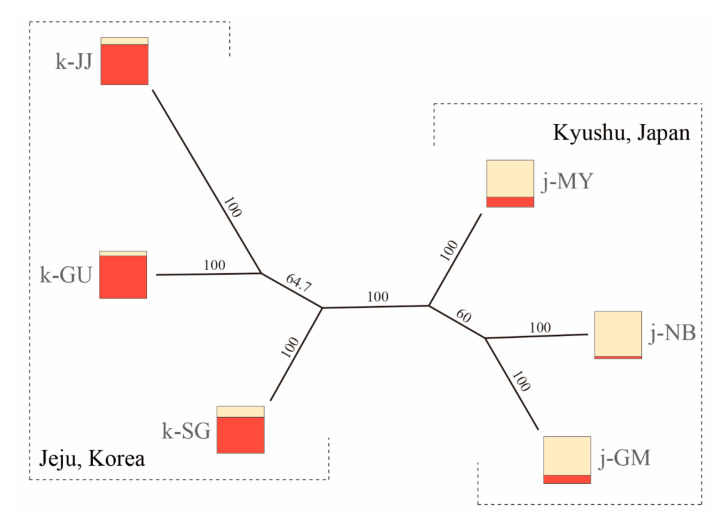
Neighbor-joining (N-J) tree based on *F*_st_ genetic distance among populations. Figures in tree branches are percentage bootstrap values estimated from 1000 reiterations. The square marks indicate the overall genotype assignment for each population to particular genetic clusters based on STRUCTURE analysis.

**Figure 5 genes-11-01114-f005:**
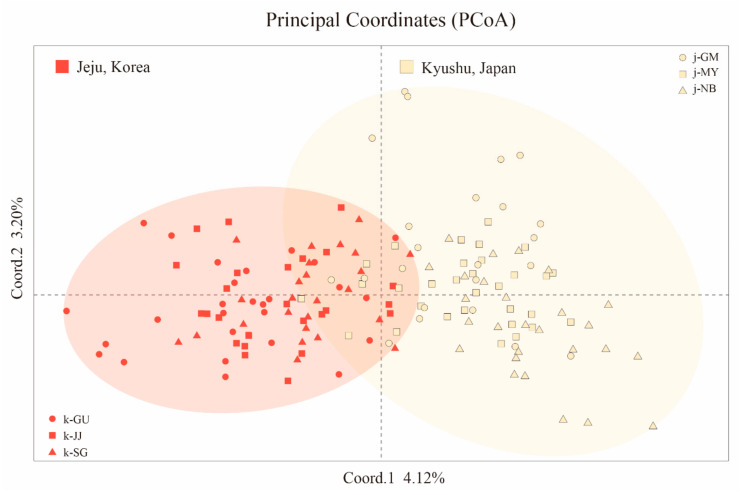
Principal coordinates analysis based on Nei’s genetic distance calculated from the allele frequencies of the 158 individuals for *Quercus gilva*. The orange symbols indicate individuals of Jeju region, and the yellow ones indicate Kyushu region. Six subgroups indicate each populations of *Q. gilva*.

**Figure 6 genes-11-01114-f006:**
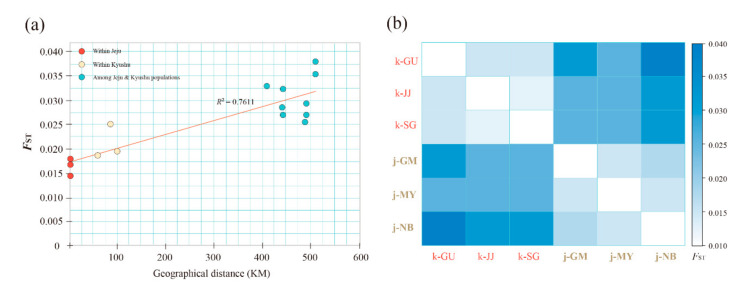
The genetic differentiation for the six populations of *Quercus gilva*. (**a**) Mantel tests between *F_ST_* values and geographical distance among populations; (**b**) distance matrix of pairwise *F_ST_* between populations.

**Figure 7 genes-11-01114-f007:**
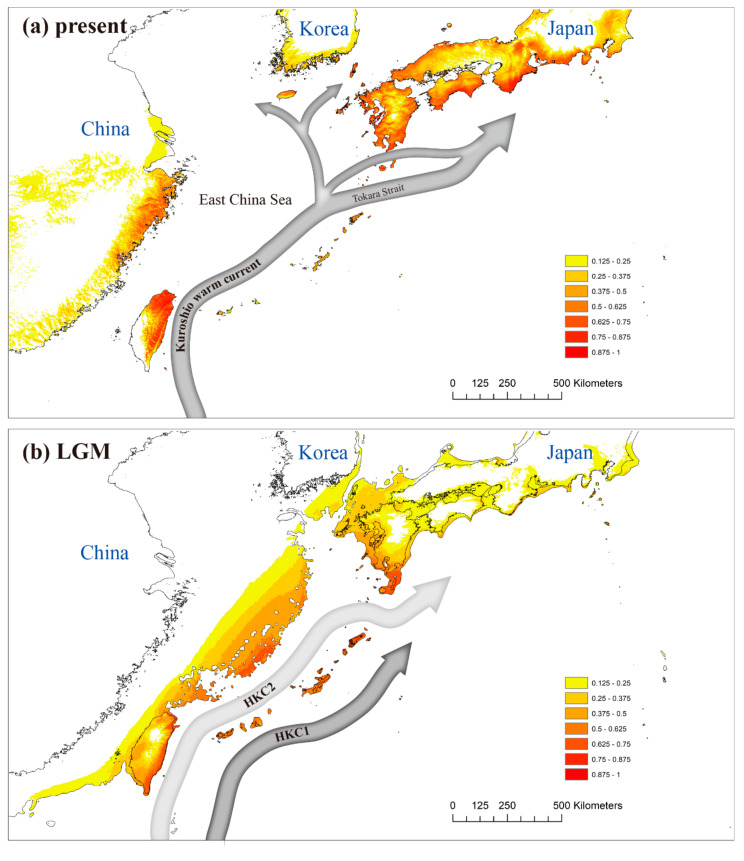
Potential distributions of *Quercus gilva* during (**a**) the present; (**b**) the Last Glacial Maximum, LGM. Distributions predicted by ecological niche modeling; potential distribution during the LGM was averaged from three general circulation models. HKC 1 indicates the main track of the Kuroshio Current during the LGM proposed by Ujiie et al. [66], Kao et al. [67], and Zheng et al. [68]. HKC 2 is the hypothetical Kuroshio Current during the LGM suggested by Vogt-Vincent and Mitarai [69].

**Table 1 genes-11-01114-t001:** Characterization of six multiplexes of 46 microsatellite loci for *Quercus gilva.*

Locus	Primer Sequence (5′–3′)	Repeat Motif	Number of Alleles	Size Range (bp)	Fluorescent Label	GenBank Accession No.
Multiplex mix A						
Qrg001	F: TCTGATGAGGTGCTGGAA R: TTGTTATCCAATTCTCTCCCT	(TC)_12_	7	100–118	6-FAM	MT811115
Qrg002	F: TGAGCTTGTTGATTGGAGAA R: CTTCAAGACGTACTACAGCA	(CA)_12_	6	158–172	6-FAM	MT811116
Qrg003	F: TTGGTGGAAGAGATTGTGAG R: CTCTTTGGGTTCTCTGTTGT	(CT)_14_	7	213–225	6-FAM	MT811117
Qrg004	F: TGGCTTCCTGACCATACATA R: GACTAACCCTGCCCTCAA	(GAA)_6_	6	107–122	VIC	MT811118
Qrg006	F: CTCAATGGCGAAATCATCAG R: TCTATAGAGGCAGCAAACAC	(TTAG)_8_	5	220–236	VIC	MT811119
Qrg007	F: GTTGGATTGGATTCTGTTGC R: TTCCCTCCTTGTCACGTT	(AG)_12_	15	103–135	NED	MT811120
Qrg008	F: ATCGGAGCAAGAAATCAAAT R: CCACCAACTCTAATGCTGTA	(AAG)_8_	3	159–168	NED	MT811121
Qrg009	F: CACTCTCTTCGACCTTCTTT R: TTCTGGGTTCTTGCTTATCG	(TCA)_9_	6	225–240	NED	MT811122
Multiplex mix B						
Qrg011	F: CGTTCAGATCAGGGTACAAA R: ATAAGCAAAGCACCCATGTA	(CA)_14_	5	160–170	6-FAM	MT811123
Qrg012	F: ATTAATGGAGAACTGCCCTC R: AGGATCATGAACTTCGACTG	(CTT)_11_	5	223–235	6-FAM	MT811124
Qrg013	F: TCTCAAACGGACCCATTTAA R: TCCTGTGATTACTGTCTATGC	(CT)_13_	5	108–120	VIC	MT811125
Qrg014	F: GTCAGTATAGCATGTGGTGT R: TTGGTGAGTTGAGATTGCAA	(GA)_14_	8	159–189	VIC	MT811126
Qrg015	F: TTCCCATTTCAGACAAGAGG R: GATTCGAACCCTCCTACAAA	(TAAC)_7_	7	209–237	VIC	MT811127
Qrg016	F: CTCTACCATCAACATCCTGC R: AATTCCAGTTTTGCAGTCCA	(AGAC)_6_	6	124–148	NED	MT811128
Qrg017	F: ACACCAAACAAAGCAAACAA R: TACGAACACAATCCAAACCT	(AACA)_6_	3	163–171	NED	MT811129
Qrg018	F: CAACCACAATGTGTAAAGACA R: GCAAAAGAGTGTATGTGCTC	(ACA)_10_	4	218–236	NED	MT811130
Multiplex mix C						
Qrg019	F: AACTCTTGCTCCATTCATTT R: GGGTCTACAATTGAATTATGGC	(AG)_13_	8	133–149	6-FAM	MT811131
Qrg020	F: AGGATTTGTAGCTGACCCTA R: GCCAAGTAATCAAATTGACTGA	(GTT)_8_	4	166–178	6-FAM	MT811132
Qrg021	F: ACAAAGACTACGTGTGCATA R: TTTCTATGAAACGCAACAGC	(CT)_14_	10	229–253	6-FAM	MT811133
Qrg022	F: GGATGACATGGCTGATCTTC R: ATAACTGGAATGGCATGGAG	(AAG)_7_	3	123–135	VIC	MT811134
Qrg024	F: CCTAAGAAGCACAGGTAAGG R: AGAGCAAGTGAGAAAGAGTC	(CT)_14_	11	237–263	VIC	MT811135
Qrg025	F: CATATAGCCGAGGAAGAAGT R: GAAGGCAGAGGTTGGTTAAA	(GAA)_6_	2	134–137	NED	MT811136
Qrg026	F: GATGGGAATGCTCTTAGGTC R: TTGTGAAGTCGCCTACAATT	(ATAG)_6_	3	180–188	NED	MT811137
Qrg027	F: TGGAAATGACATTGTTACCCT R: CCGATGACAAGAATCCCAAT	(GA)_14_	12	235–271	NED	MT811138
Multiplex mix D						
Qrg028	F: TAAAGGAGTGCATGGTGAAA R: AGTGAAGCCTCTTTCCTAGA	(CT)_13_	9	127–147	6-FAM	MT811139
Qrg029	F: AAGATAACTGCACGCTTGTA R: TCAGAAATCGCTCATACCTG	(TG)_13_	7	184–196	6-FAM	MT811140
Qrg030	F: CTATTCATGGACTCCTCTGT R: AATTGCAAGGCCTTAGAACT	(AG)_15_	7	235–249	6-FAM	MT811141
Qrg031	F: GGTTAGGGCTCTTTCCAAAT R: CTCTCCCTTTCTTTCACTGT	(GA)_13_	8	131–145	VIC	MT811142
Qrg033	F: TCTTGCCAATCTAAATCCCA R: TGCATGATACAGAAACACCA	(AAGA)_7_	2	239–247	VIC	MT811143
Qrg034	F: GGACATCTACAGCCTACAAA R: CGCAGACCAAATATCATTCTC	(CT)_12_	12	143–173	NED	MT811144
Qrg036	F: TAACTTTGTTCTCGCCTGA R: AATGTAGAGCCTGTTTGCAT	(GA)_13_	7	239–259	NED	MT811145
Multiplex mix E						
Qrg037	F: TTCGAGATAGGACAGAGGAG R: TGTGTTTGATTAGCGGAGAA	(AAGA)_8_	5	128–144	6-FAM	MT811146
Qrg038	F: TGGCTATGATAATTGTGGGT R: CTCAACCCTGTTATCTCACC	(GA)_17_	8	182–204	6-FAM	MT811147
Qrg039	F: AAAGTGGATTTGCAGCCTAA R: GACAATGGAGAAGGGACAAT	(TC)_14_	6	244–260	6-FAM	MT811148
Qrg040	F: GCATTTCTCTCTCTGGTTCA R: AAGTACCCTCCATCTACGTT	(AAG)_6_	3	128–146	VIC	MT811149
Qrg041	F: CTTCCTCGTCAATAGTCCAC R: AGTGAGTTTGATACGCTTGT	(AAG)_12_	9	186–228	VIC	MT811150
Qrg042	F: CCCACACATTATACCACGAA R: CTACTAACAACCGCAACTCT	(AG)_17_	8	227–253	VIC	MT811151
Qrg043	F: CATACATCCTAGTGCAGCAG R: GGTAGCTCAAGTTCACAGTT	(CAA)_6_	2	149–155	NED	MT811152
Multiplex mix F						
Qrg046	F: CTGCCCCTAACTAATCTGTT R: GTAGATGATGAGGTTGTGGG	(TGT)_6_	2	149–152	6-FAM	MT811153
Qrg047	F: AGACCAGTAGATGCTTCAAA R: ATTCATGACCCTCCTTCTCA	(AAG)_9_	3	208–217	6-FAM	MT811154
Qrg048	F: TCCATCGTCAACAAAGGATT R: AACCAGTTCTCACTCTCTCT	(AG)_17_	7	235–269	6-FAM	MT811155
Qrg049	F: CAACTACTGTAGCCTTGTGT R: TATGCCTCCAGTGTACTACA	(CA)_12_	7	146–166	VIC	MT811156
Qrg050	F: GGGACCATAGCAGTGTTAAT R: AGCCCTCCCTTATTTATTCC	(TC)_21_	8	192–216	VIC	MT811157
Qrg051	F: CTCCTCTTGGCTATGACATC R: TCTTGTTTGAGGAAGTTGACA	(TTC)_14_	10	235–259	VIC	MT811158
Qrg052	F: ACTTGTAACTAACCTGGCTC R: CTAGGAGGATGAAATGGCAA	(CTAA)_8_	4	150–162	NED	MT811159
Qrg053	F: TGACAGTACATGGTAAAGCT R: TTCTTGGTCTTGAATGAGGA	(CT)_14_	7	204–228	NED	MT811160

**Table 2 genes-11-01114-t002:** Genetic parameters for 46 microsatellite loci across all samples developed for *Quercus gilva.*

Locus	*N_A_*	*H* _O_	*H* _E_	Null	Locus	*N_A_*	*H* _O_	*H* _E_	Null
Qrg001	8	0.703	0.619	0.0020	Qrg027	16	0.829	0.865	0.0031
Qrg002	10	0.741	0.755	0.0053	Qrg028	10	0.842	0.820	0.0018
Qrg003	7	0.741	0.794	0.0112	Qrg029	8	0.677	0.743	0.0275
Qrg004	7	0.778	0.786	0.0057	Qrg030	9	0.253	0.765	0.2774 *
Qrg006	6	0.772	0.763	0.0033	Qrg031	10	0.810	0.841	0.0091
Qrg007	18	0.918	0.899	0.0012	Qrg033	3	0.222	0.209	0.0048
Qrg008	4	0.361	0.366	0.0058	Qrg034	16	0.867	0.885	0.0038
Qrg009	8	0.570	0.748	0.0868 *	Qrg036	9	0.627	0.800	0.0910 *
Qrg011	6	0.684	0.745	0.0071	Qrg037	5	0.696	0.738	0.0118
Qrg012	7	0.759	0.756	0.0026	Qrg038	13	0.759	0.794	0.0076
Qrg013	8	0.513	0.657	0.0728 *	Qrg039	6	0.753	0.731	0.0035
Qrg014	11	0.861	0.835	0.0016	Qrg040	3	0.468	0.539	0.0218
Qrg015	7	0.620	0.671	0.0073	Qrg041	13	0.823	0.808	0.0025
Qrg016	6	0.608	0.622	0.0044	Qrg042	14	0.734	0.805	0.0204
Qrg017	5	0.589	0.597	0.0117	Qrg043	3	0.133	0.131	0.0058
Qrg018	6	0.532	0.631	0.0464	Qrg046	3	0.348	0.384	0.0192
Qrg019	10	0.759	0.824	0.0121	Qrg047	4	0.475	0.496	0.0067
Qrg020	5	0.633	0.674	0.0050	Qrg048	12	0.595	0.808	0.0977 *
Qrg021	19	0.861	0.886	0.0062	Qrg049	10	0.430	0.505	0.0353
Qrg022	3	0.165	0.179	0.0099	Qrg050	14	0.791	0.819	0.0044
Qrg024	13	0.741	0.857	0.0355	Qrg051	10	0.810	0.814	0.0036
Qrg025	2	0.044	0.067	0.0316	Qrg052	4	0.551	0.572	0.0106
Qrg026	4	0.139	0.203	0.0607 *	Qrg053	10	0.741	0.756	0.0029

*N_A_*, number of alleles; *H*_O_, observed heterozygosity number of alleles; *H*_E_*,* expected heterozygosity; Null, null allele frequency estimate. * indicates that the frequency of the null allele exceeds 5%.

**Table 3 genes-11-01114-t003:** Summary statistics of genetic diversity for six populations based on 40 microsatellite loci of *Quercus gilva.*

ID	Location	Coordinates	*E* _V_	*N*	*N* _A_	*A* _R_	*P* _A_	*P* _riv_	*H*_O_ (SE)	*H*_E_ (SE)	*F* _IS_
Jeju Island											
k-GU	Gueok-ri, Daejeong-eup, Jeju	33°18′8.21″ N, 126°16′36.59″ E	136	32	254	5.729	11	0.043	0.660 (0.037)	0.645 (0.033)	− 0.024
k-JJ	Jeoji-ri, Hangyeong-myeon, Jeju	33°18′45.36″ N, 126°17′3.76″ E	168	27	254	5.905	4	0.016	0.656 (0.040)	0.640 (0.035)	0.003
k-SG	Seogwang-ri, Andeok-myeon, Jeju	33°17′57.47″ N, 126°18′59.97″ E	201	18	237	5.925	8	0.034	0.639 (0.039)	0.651 (0.033)	0.043
Mean			168.3	25.6	248.3	5.853	7.7	0.031	0.652	0.645	0.007
Pooled populations				77	301	7.525	33	0.110	0.641	0.657	0.018
Kyushu											
j-GM	Kuma-gun, Kumamoto Prefecture	32°17′39.5″ N, 130°52′17.6″ E	485	23	238	5.656	5	0.021	0.638 (0.039)	0.634 (0.036)	−0.005
j-MY	Miyakonojo-shi, Miyazaki Prefecture	31°50′55.7″ N, 131°13′30.4″ E	230	29	261	5.940	10	0.038	0.636 (0.034)	0.647 (0.035)	0.003
j-NB	Nobeoka-shi, Miyazaki Prefecture	32°39′15.8″ N, 131°41′14.3″ E	38	29	224	5.201	4	0.018	0.613 (0.043)	0.615 (0.038)	0.023
Mean			251	27	241	5.600	6.3	0.026	0.629	0.632	0.007
Pooled populations				81	302	7.503	34	0.113	0.628	0.648	0.015

*E*_V_, elevation of sampling site (meter); *N*, number of individuals; *N*_A_, number of alleles; *A*_R_*,* allelic richness; *P*_A_*,* number of private alleles; *P*_riv_, private allelic rate; *H*_O_, observed heterozygosity number of alleles; *H*_E_*,* expected heterozygosity; SE, standard error; *F*_IS_, inbreeding coefficient.

**Table 4 genes-11-01114-t004:** Probability of a bottleneck estimated using the program BOTTLENECK for six populations of *Quercus gilva,* based on the two-phase model (TPM) or stepwise mutation model (SMM).

Population	Wilcoxon Test		Mode Shift
	TPM	SMM	
Jeju Island			
k-GU	0.476153	0.988818	No
k-JJ	0.465576	0.991747	No
k-SG	0.294988	0.925049	No
Kyushu			
j-GM	0.383349	0.974584	No
j-MY	0.292696	0.982035	No
j-NB	0.135276	0.765987	No

## Data Availability

Sequence data for microsatellite loci developed in this study are available on GenBank (accession numbers MT811115–MT811160).

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
