# Peer review of "A Disjunctive Marginal Edge of Evergreen Broad-Leaved Oak (Quercus gilva) in East Asia: The High Genetic Distinctiveness and Unusual Diversity of Jeju Island Populations and Insight into a Massive, Independent Postglacial Colonization"

_genes, 2020, doi:10.3390/genes11101114_

Round 1

Reviewer 1 Report

“A distinctive marginal edge of evergreen broad-leaved oak (Quercus gilva) in East Asia: The high genetic distinctiveness and unusual diversity of Jeju Island populations and insight into a massive, independent postglacial colonization”

Reviewed for Genes

This paper describes a systematic analysis of an intriguing question, on the origin and uniqueness of an island population of Quercus gilva that is at the edge of the species’ distribution and of potential conservation concern.  The authors incorporated a variety of genetic techniques to measure diversity, screen for past bottlenecking, and document current structure, as well as environmental modeling techniques to map the past and current species distribution.  Overall, this is a clear, well-organized and well-written paper, though I have a few questions and suggestions. 

First, the authors appear to have taken great care in identifying and quantifying the microsatellite loci, to the point of correcting for potential biases from null alleles, and in the subsequent statistical analyses.  They clearly demonstrate that the populations cluster into two groups, corresponding to those in Jeju and Kyushu, that the two regions have comparable levels of genetic variation, and that neither is ancestral to the other.  I note that the Mantel test correlating Fst values with geographic distances between the Jeju and Kyosho populations is over water, even apparently during the Last Glacial Maximum (LGM) (from Fig. 2).  Distances over land and water are not generally considered comparable, and the authors should address whether this changes their interpretations.  I don’t think it would.

Second, some scientists distinguish between leading and trailing edge populations during range shifts, with both exhibiting lower genetic diversity but for different reasons and of different kinds (e.g., founder effects, relict populations).   Is this distinction relevant to the current paper and, if so, would it change any of the interpretations?

Third, I would like to see justification for the use of the term “massive” in referring to the colonization of Jeju.   Clearly this small island population does not have the low levels of genetic variation expected from founder effects.  Instead, it has comparable levels of genetic variation, by many measures, as the larger population on Kyushu, but we don’t know how much larger that population is.  (Is the Jeju count of 600 individuals just of canopy trees or does it include saplings, etc?  Also, can the authors estimate the effective population sizes from their genetic data?  This might warrant a second manuscript but would be worth mentioning here).  More importantly, we are not told if these shared levels of genetic diversity are high or perhaps typical for oak species in the region.  I may well be missing something, but the reference to “massive, independent postglacial colonization” seems to warrant further explanation.

Fourth, while the Jeju populations clearly differ from those on Kyushu and are of independent origin, that by itself would not seem to justify special conservation attention.  We would need to know whether the Jeju populations are also different from other populations in East Asia, some perhaps derived from the same ancestral populations.  Again, the authors may have indirectly supplied this information, but they should make it explicit.

Finally, the authors use Ecological Niche Modeling to fit the distribution of the species to climate variables and elevation both in the present and during the Last Glacial Maximum.  This is entirely appropriate and they follow common practices.  Climate is tightly linked with elevation, and LGM elevations were apparently taken from “CHELSEA”, which is not referenced.  I would like to see a reference to this source, and a statement as to whether it corrects for isostatic effects, in which northern land masses sink and rise with the expansion and then disappearance of the ice masses.  This can change the amount and location of land areas at various heights above sea level and should be considered in any such analysis.  After a quick search in the literature I found one article that suggested that some areas in Japan experienced a 30 meter rebound.  This number might vary, and may be too small to affect the conclusions, but I the issue should be addressed.

Finally, on line 101 I propose changing the phrase “management of the threatened species” to “management of threatened populations” as the species as a whole appears not to be at risk.

Reviewer 2 Report

In this MS, the authors developed SSR markers for the Quercus gilva and used the marker to analyze the genetic diversity of 158 genotypes from 6 locations. The paper was well organized, but some details authors may want to check and correct before accepting.  

In abstract: authors may want to show the full name of the PCoA

In row 194-195, "Two occurrence points of Korean Peninsula and Toyama Prefecture in Japan were excluded because they were assumed to be inaccurate.", authors may need to provide the reasons that they assume the data is not accurate.

In row 214: "300 × 300 base pairs", what authors want to show?

In row 252-255: The authors may need to clarify what the abbreviation means, such as j-NB, k-GU, k-SG, k-JJ, and j-MY.

For the figure order, I felt the order of the figure not match our thinking. Such as Figure 4a,b should be before Figure 3a. Because figure 3a or even figure 2 was based on the result figure 4a,b.

For Figure 5. Could the authors marker the six subgroups based on sampling locations? It might be interesting to show the differentiation among them.  
